*Method*

# CODEX, a neural network approach to explore signaling dynamics landscapes

Marc-Antoine Jacques[1] ID, Maciej Dobrzyński[1] ID, Paolo Armando Gagliardi[1] ID, Raphael Sznitman[2] & Olivier Pertz[1,*] ID

## Abstract

Current studies of cell signaling dynamics that use live cell fluorescent biosensors routinely yield thousands of single-cell, heterogeneous, multi-dimensional trajectories. Typically, the extraction of relevant information from time series data relies on predefined, human-interpretable features. Without a priori knowledge of the system, the predefined features may fail to cover the entire spectrum of dynamics. Here we present CODEX, a data-driven approach based on convolutional neural networks (CNNs) that identifies patterns in time series. It does not require a priori information about the biological system and the insights into the data are built through explanations of the CNNs' predictions. CODEX provides several views of the data: visualization of all the single-cell trajectories in a low-dimensional space, identification of prototypic trajectories, and extraction of distinctive motifs. We demonstrate how CODEX can provide new insights into ERK and Akt signaling in response to various growth factors, and we recapitulate findings in p53 and TGFβ-SMAD2 signaling.

**Keywords** cell signaling; convolutional neural network; data exploration; live biosensor imaging; time series analysis

**Subject Categories** Computational Biology; Methods & Resources; Signal Transduction

**Mol Syst Biol. (2021) 17: e10026**

## Introduction

Cell signaling dynamics rather than steady states control fate decisions (Purvis & Lahav, 2013). Biosensor imaging has documented heterogeneous and asynchronous p53 (Lahav *et al,* 2004), NF-kB (Tay *et al,* 2010), ERK (Albeck *et al,* 2013; Ryu *et al,* 2015), Akt (Sampattavanich *et al,* 2018), and SMAD (Strasen *et al,* 2018) signaling dynamics across individual cells of a population. This heterogeneity arises both from biological noise extrinsic to individual cells and from intrinsic variability within signaling networks (Snijder & Pelkmans, 2011). Current biosensor measurements usually yield up to thousands, potentially multivariate signaling trajectories (Sampattavanich *et al,* 2018). The characterization of these trajectories typically relies on visual inspection, followed by statistical analysis of hand-crafted, human-interpretable features (Gillies *et al,* 2017; Sampattavanich *et al,* 2018). The volume and high dimensionality of these datasets, as well as the capacity of features to faithfully describe complex dynamics of the trajectories make data mining challenging. To meet these challenges, we present CODEX (COnvolutional neural networks for Dynamics EXploration), a data-driven approach for the exploration of dynamic signaling trajectories using convolutional neural networks (CNNs). It benefits from the ability of CNN classifiers to identify a set of data-driven features that best summarizes the data. CODEX turns the information learnt by CNNs into three complementary views of the data: A low-dimensional representation of the whole dataset that emphasizes the distribution of signaling dynamics, a set of prototypical time series, and a collection of signature motifs. Notably, CODEX regroups all these results in a single framework. This enables to quickly obtain a complete overview of the dataset and to interactively combine the results to form rich visualizations of signaling dynamics.

## Results

In a typical CODEX analysis, a CNN classifier is trained to recognize single cells from various experimental conditions (the input classes) based on their corresponding time series (Fig 1A). Any time series behavior that constitutes a distinctive feature (e.g., repeated motifs, trends, or baselines) is henceforth referred to as dynamics. To avoid the difficulties related to CNN training, we use a simple, yet powerful CNN architecture (Appendix Note 1, Table EV1). This architecture builds a compressed representation of the input, hereafter referred to as CNN features, which is a one-dimensional vector used for classification. We found that this architecture was simple to adapt to a range of datasets because reducing the number of CNN features provides an easy and sufficient way to counter overfitting. Although there are more parameters that impact the training process (e.g., L2 norm), we found the architecture to perform robustly with default values across multiple time series datasets (see Materials and Methods).

1 Institute of Cell Biology, University of Bern, Bern, Switzerland
2 ARTORG Center for Biomedical Engineering Research, University of Bern, Bern, Switzerland
 *Corresponding author. Tel: +41 31 631 46 37; E-mail: olivier.pertz@izb.unibe.ch

## CODEX extracts data-driven features that isolate and expose dynamics in synthetic data

We first demonstrate CODEX using synthetic trajectories that mimic the pulsating signal typically observed for different signaling pathways in cells (Albeck *et al*, 2013; Stewart-Ornstein & Lahav, 2017) (see Materials and Methods, Fig EV1A, Appendix Note 2). All synthetic trajectories display four peaks that can be of two types: either a full or a truncated Gaussian peak. The trajectories are split into two classes: In the first class, trajectories have a majority of full peaks (i.e., 2, 3, or 4 full peaks), while in the second class they have a majority of truncated peaks (i.e., 0, 1, or 2 full peaks). This dataset was constructed such that only the abundance of full peaks can be used to separate one class from another. The trajectory baseline is a non-discriminative feature, and neither the timing of the peaks nor the order of the peaks can be used as these are randomly sampled. In this setup, we trained a CNN that quickly converged to the optimal accuracy

(i.e., $5/6 \approx 83\%$, because of the ambiguous case where a trajectory has two full peaks).

A t-distributed stochastic neighbor embedding (t-SNE) projection of the CNN features revealed that the CNN learnt to distinguish the classes by recognizing and counting the different types of pulses (Fig EV1B). This demonstrates that despite the CNN being solely trained to separate the input classes, its latent representations of the input can be used to reveal structures in the data which are linked to specific dynamics. We then showed that this feature space can be sampled to identify prototype trajectories representative of different dynamics (Fig EV1C).

Further, we used class activation maps (CAMs) (preprint: Zhou *et al*, 2015) to identify class-specific motifs. CAMs are based on the so-called "model attention" to highlight parts of the input that are important for the CNN to recognize a given class. By building CAMs for many input trajectories, we extracted a collection of signature motifs for each class (Fig EV1D, Movie EV1). Subsequently, we clustered the motifs using the dynamic time warping (DTW)

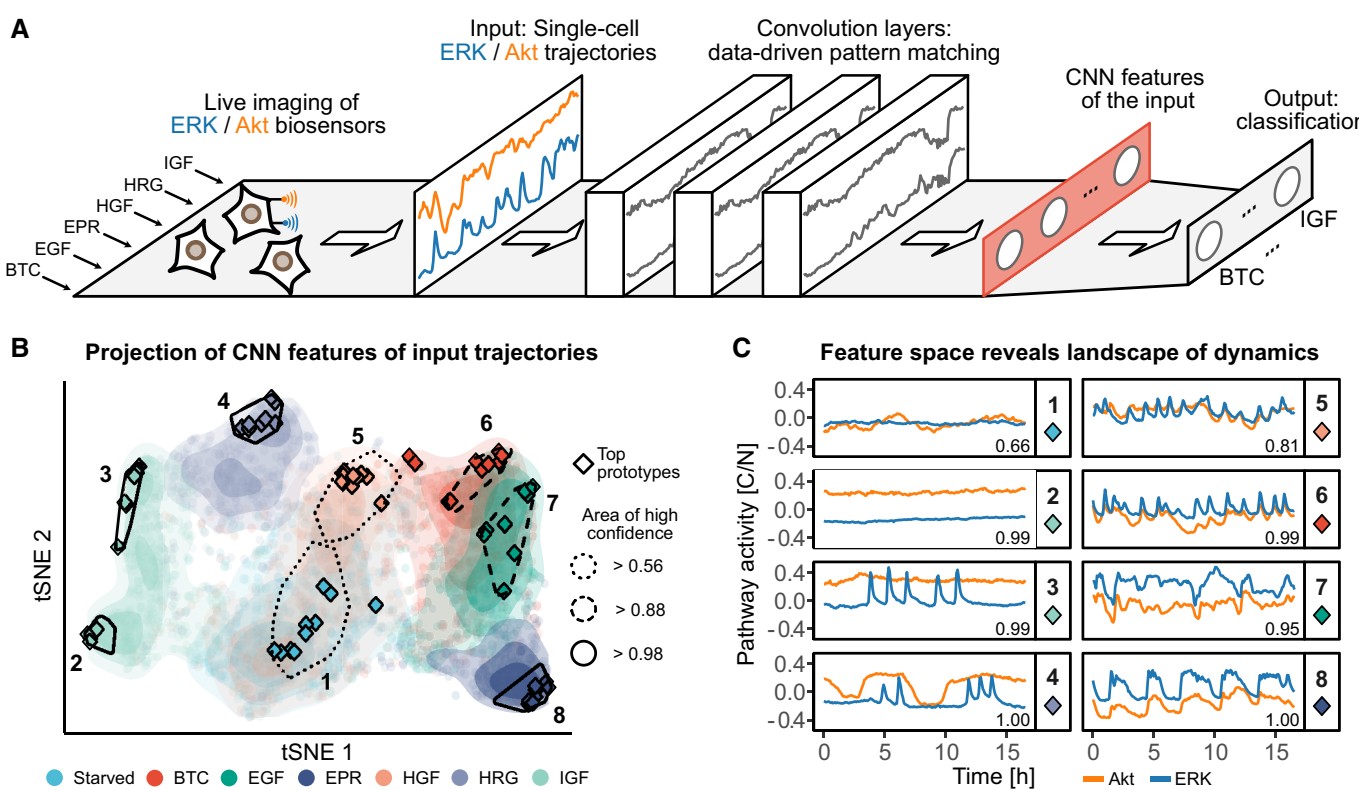

**Figure 1. CNN latent features reveal the landscape of single-cell signaling dynamics.**

A   Schematic of the CNN classifier architecture used in CODEX. In this example, MCF10A cells are exposed to 6 different GFs and single-cell ERK/Akt activity is reported using biosensors. The GF treatments form classes that the classifier is trained to identify based on the bivariate ERK/Akt single-cell input trajectories. The input trajectories are passed through a cascade of convolution layers, followed by a global average pooling layer that compresses the convolution responses into a one-dimensional vector of features (red rectangle). This latent representation of the input is then referred to as CNN features. Finally, the CNN features are passed to a fully connected layer to perform the classification.

B   t-SNE projection of the CNN features of all ERK/Akt trajectories in the validation set. Each point represents a bivariate ERK/Akt trajectory from a single cell. Hulls indicate areas associated with a strong classification confidence for each GF. Shading shows the point densities. Diamonds indicate the positions of the 10 top prototypes (see Materials and Methods) for each GF. The solidity of the hull contour line indicates the minimal classification confidence for all prototypes in the hull.

C   Representative ERK/Akt prototype trajectories from each hull indicated in (B). Pathway activity is reported as the ratio of cytosolic over nuclear KTR fluorescence (C/N). Each pathway activity was preprocessed by removing the average activity of this pathway across all trajectories in the training set. Numbers in the bottom-right corners indicate the CNN predicted probability for the trajectories to belong to their actual class.

distance to visualize the diversity of motifs (see Materials and Methods, Fig EV1D and E). This shows that CAMs exhaustively extracted all the different motifs from the dataset, in a class-specific manner.

## Mapping the signaling dynamics landscape of ERK and Akt induced by various growth factors

After this proof of concept, we applied CODEX to study Ras-ERK/ PI3K-Akt single-cell signaling dynamics. The activity of both ERK and Akt was measured using a multiplexed genetically encoded biosensor system and quantified using a computer vision pipeline (Fig EV2A and B). The system relies on kinase translocation reporter (KTR) fluorescent biosensors (Regot *et al,* 2014). These sensors provide a readout of the associated pathways activity through ratio of cytosolic over nuclear KTR fluorescence intensity (C/N). To induce distinct ERK/Akt dynamics, we stimulated MCF10A breast epithelial cells with different growth factors (GFs) (Sampattavanich *et al,* 2018). Cells were treated with 100 ng/ml with any of the following GFs: epidermal GF (EGF), betacellulin (BTC), epiregulin (EPR), hepatocyte GF (HGF), heregulinβ-1 (HRG), and insulin-like GF 1 (IGF). An additional class consisted of starved cells that were left untreated. For each GF, we acquired at least 1,200 single-cell, bivariate ERK/Akt trajectories across two replicates for 48 h. In the $1^{st}$ phase, lasting about 8 h, population-synchronous dynamics were observed due to acute GF stimulation (Fig EV2C–E). This was particularly visible when evaluating cell population-averaged measurements (Fig EV2E). This was followed by a $2^{nd}$ phase characterized by heterogeneous, asynchronous dynamics that cannot be captured by a population average. The latter dynamics were shown to be relevant for proliferation fate decisions (Albeck *et al,* 2013; Sampattavanich *et al,* 2018), but are difficult to analyze with traditional methods because of their asynchrony and heterogeneity. For the rest of our analysis, we truncated the trajectories to keep only the $2^{nd}$ phase and focus on this difficult part of the data. To explore the specific signaling dynamics induced by each GF, we trained a CNN classifier that takes bivariate ERK/Akt trajectories as input to predict starvation or treatment with a specific GF. After training, this classifier separated the different classes with about 65% accuracy (Tables EV2 and EV3), suggesting that input trajectories carry distinctive features that depend on GF identity many hours after exposure.

To map the landscape of signaling dynamics induced by the different GFs, we used a t-SNE projection of the CNN features learnt for each trajectory (Fig 1B). GFs populated well-defined but also partially overlapping areas in the projection space. Trajectories of starved cells localized to a central largely spread cluster (area 1). Left of area 1, IGF (IGFR ligand) led to a polarized cluster that we designated as areas 2 and 3, while HRG (ErbB3/4 ligand) formed a distinct cluster (area 4). HGF formed area 5 that overlapped greatly with starved cells. Right of area 1, a cluster was formed by EGF, EPR, and BTC, which are all ErbB1 ligands. This suggests that each GF induces a specific continuum of heterogeneous signaling dynamics, whose characteristics correlate with their cognate GF receptor. Finally, we noted that GFs which overlapped in the projection, corresponded to cases where the CNN classification performance was low (Table EV2).

We built a web application to interactively explore different areas of the t-SNE projection and to report their associated ERK/Akt trajectories (Movie EV1). We first evaluated trajectories that

exhibited the highest classification accuracies for each class, which we refer to as "top prototypes" (see Materials and Methods, Fig 1C). Through visual inspection, we empirically described the main trends of the prototypes using qualitative features. Starved cells displayed flat baseline ERK/Akt levels (Fig 1C, area 1). The polarized cluster induced by IGF revealed two distinct dynamics: both displayed sustained Akt activity, but area 2 displayed low ERK activity, while area 3 displayed pulsatile ERK activity. HRG induced trains of sharp ERK activity pulses enveloped by wide Akt pulses of high amplitude (area 4). HGF induced pulsatile, sharp, synchronous ERK, and Akt pulses (area 5). However, the large overlap of HGF-induced trajectories with area 1 indicates that many cells had adapted and behaved as starved cells. The ErbB1 ligands induced similar dynamics that consisted of synchronous ERK/Akt pulses (BTC—area 6, EGF—area 7, EPR—area 8) (Yarden & Pines, 2012). The relative width of the ERK/Akt pulses, however, varied from sharp (BTC) to wide (EPR) and flat (EGF). Along the left-right axis of the projection, HRG and IGF displayed higher Akt amplitudes, while BTC, EGF, and EPR displayed lower Akt amplitudes.

A limitation of examining top prototypes is that they might not faithfully reflect the heterogeneity of signaling dynamics in a class. For example, this was illustrated by the absence of top prototypes from areas where GFs overlap (Fig 1B). Therefore, we also used an alternative sampling strategy to identify prototype trajectories whose CNN features are as uncorrelated as possible (Appendix Fig S1B and C). To ensure that the selected trajectories are still representative of each class, we only considered trajectories that reached a minimal threshold of prediction confidence for their actual class. This resulted in a better coverage of trajectories in the CNN feature space, while maintaining class specificity. Visual inspection of prototype trajectories sampled with the different strategies provides a more complete picture of the salient features that characterize the individual classes. It is also of interest to identify trajectories for which the CNN prediction was wrong despite exhibiting a large confidence (Appendix Fig S1D). For example, this might help to understand classes overlap or to identify dynamics that are rarely observed outside of a given class.

The dynamic information transferred by signaling pathways often relies on local signal shapes such as the time interval between pulses (Albeck *et al,* 2013; Ryu *et al,* 2015; Sampattavanich *et al,* 2018) or the decay kinetics (Bugaj *et al,* 2018). Using CAMs, we identified minimal signaling motifs that discriminate trajectories induced by each ligand. We then used DTW shape-based clustering to evaluate their distribution in different GF classes (Fig 2A and B). This approach again provided additional intuition about GF-specific signaling dynamics: (i) The ErbB1 ligands BTC, EGF, and EPR all led to a mix of synchronous ERK/Akt pulses in which Akt amplitude was lower than ERK amplitude (clusters 2, 3), or wider ERK/Akt pulses with a very low Akt amplitude (cluster 4); (ii) HRG led to a peculiar pattern consisting of multiple sharp ERK pulses under larger Akt pulses (cluster 5); (iii) IGF led to sustained Akt and baseline ERK activity with occasional pulses (clusters 6, 7).

To further validate the overview of ERK and Akt dynamics and show that this specific CNN did not overemphasize or overlook some trajectory features, we performed the CODEX analysis with a ResNet architecture (He *et al,* 2016) instead of the plain CNN architecture (see Materials and Methods). With this new model, the dataset projection, prototypes extraction, and CAM-based motifs all

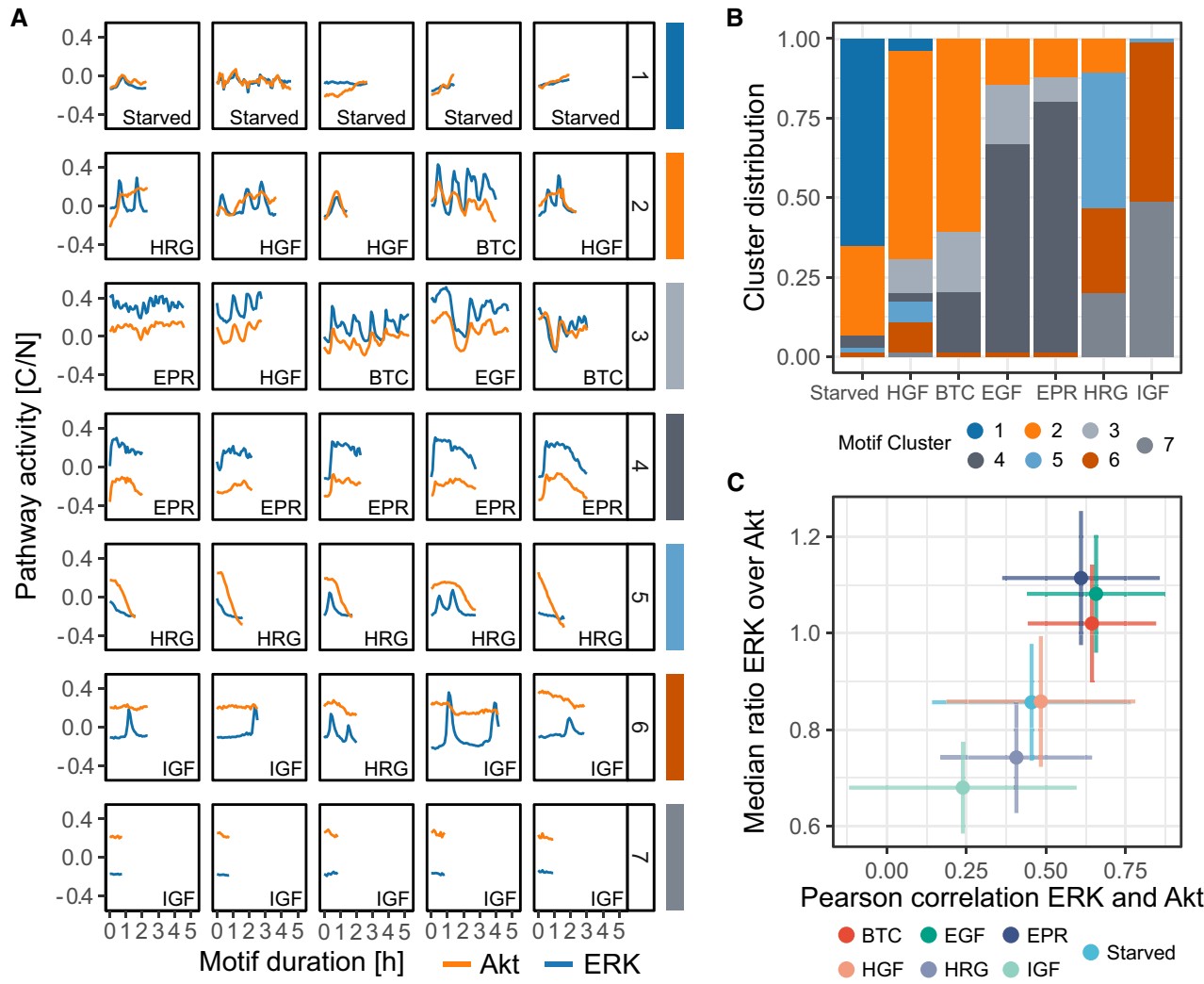

**Figure 2.  Discriminative motifs in ERK/Akt trajectories highlight GF signaling signatures and ease the identification of interpretable, GF-specific features.**

A   Discriminative signaling motifs were extracted from the training and validation top prototypes using a CAM-based approach (see Materials and Methods). The motifs were clustered using DTW distance and Ward's linkage. Representative motifs of each cluster, based on the minimization of median intra-cluster distance, are displayed (see Materials and Methods). Bottom-right labels indicate the class of the trajectory from which the motif was extracted. Each pathway activity was preprocessed by removing the average activity of this pathway across all trajectories in the training set.

B   Distribution of the signaling motifs clusters across the GF treatments.

C   Scatter plot of the Pearson correlation coefficient between ERK and Akt trajectories against the median ratio of ERK over Akt activity in single cells. For each trajectory, ratios are computed on raw data, at each time point and summarized with median. Crosses indicate the mean values and the standard deviations of all raw single-cell trajectories. At least 1,200 cells for each GF pooled from two technical replicates.

provided a picture consistent with the previous results (Fig EV3). This result indicates that the choice of the CNN architecture is flexible and that CODEX results are robust to variations in the model training step.

The insights brought by the different components of CODEX provide an intuitive picture of important dynamics that are underlying a dataset. To summarize the findings and check that the intuition built from CODEX's output was correct, a targeted extraction of explicit features is a fitted next step. For example, CODEX results suggested that the synchrony between ERK and Akt as well as their amplitude ratio were highly discriminative across GF stimulations.

We verified this by computing the temporal correlation between ERK and Akt as well as the median ratio of ERK over Akt (Fig 2C). The frequency of ERK and Akt pulses and their synchrony were also suggested and validated as discriminative features (Fig EV4).

## CODEX recapitulates the findings of classic machine-learning workflows but enhances them with a focus on dynamics motifs

To compare CODEX results against classic machine-learning approaches, we used an existing library that comprehensively extracts hundreds of explicit time series features (Christ et al, 2018)

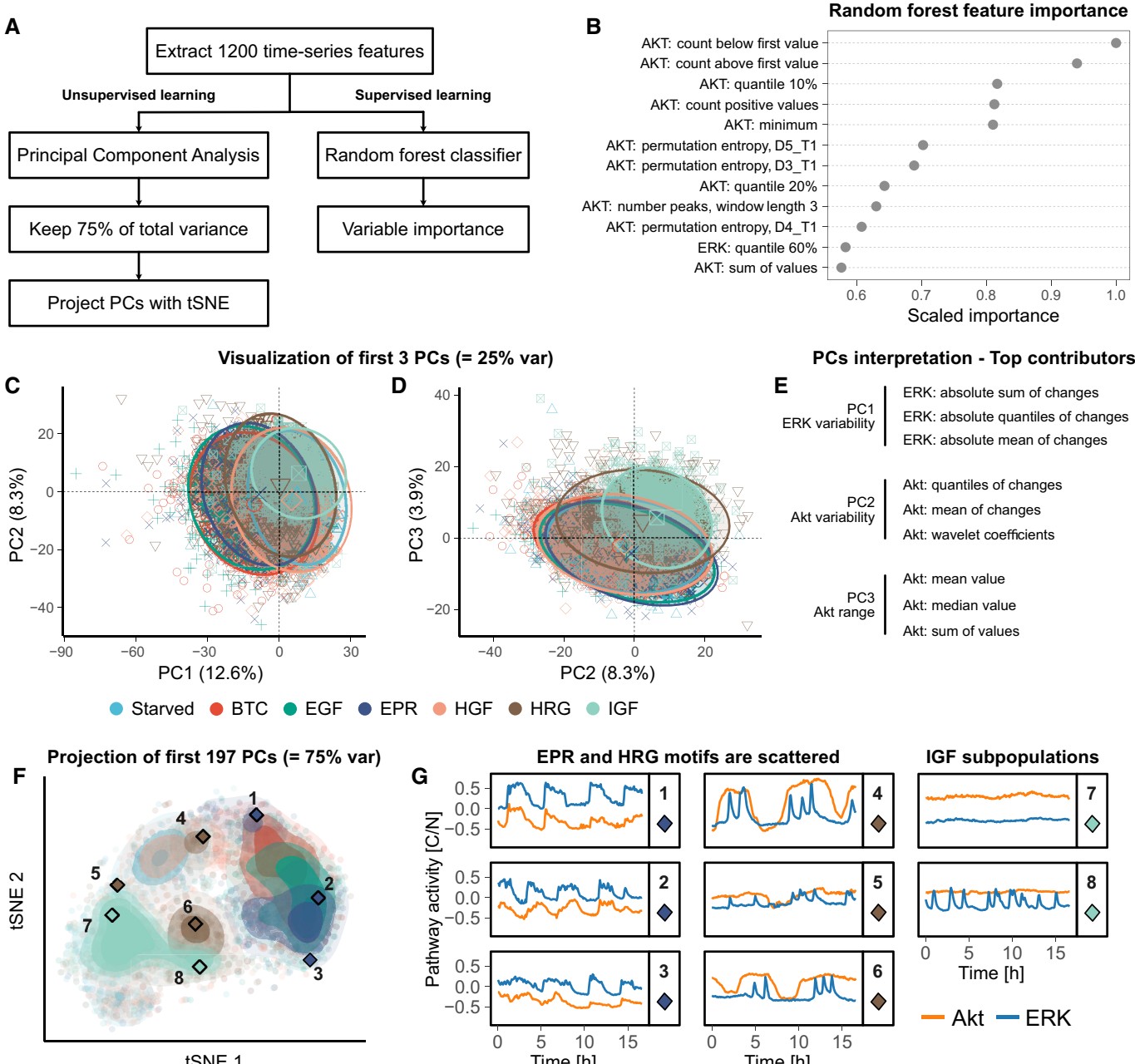

**Figure 3.  Classic machine-learning workflows for time series exploration provide a more coarse-grained overview of ERK/Akt signaling landscape than CODEX.**

A       Schematic of the two workflows used for analyzing ERK/Akt signaling in response to GFs.

B       Feature importance of a random forest trained to classify ERK/Akt time series according to the GF treatments. The 12 most important features are shown. The signaling pathway (ERK or Akt) associated with each feature is indicated. For the permutation entropy, the parameters are as follows: D length of the subwindows and T lag between the windows (Christ *et al*, 2018).

C, D    Biplots of the first three principal components derived from the time series features PCA. Each symbol represents an individual trajectory. The circles indicate normal ellipses for each group with a confidence of 95%. The larger symbols in the middle of the ellipses are visual helps for the identification of the groups. The percentiles in the axis labels indicate the amount of total variance carried by the principal components.

E       Features that contributed the most to the three first principal components (see Materials and Methods).

F       t-SNE projection of the first 197 principal components, which carry 75% of the total variance of the features. Each point represents an individual trajectory; shading indicates point density. The diamond symbols indicate the trajectories shown in (G).

G       Manually selected trajectories which are highlighted with a diamond symbol and a label in (F).

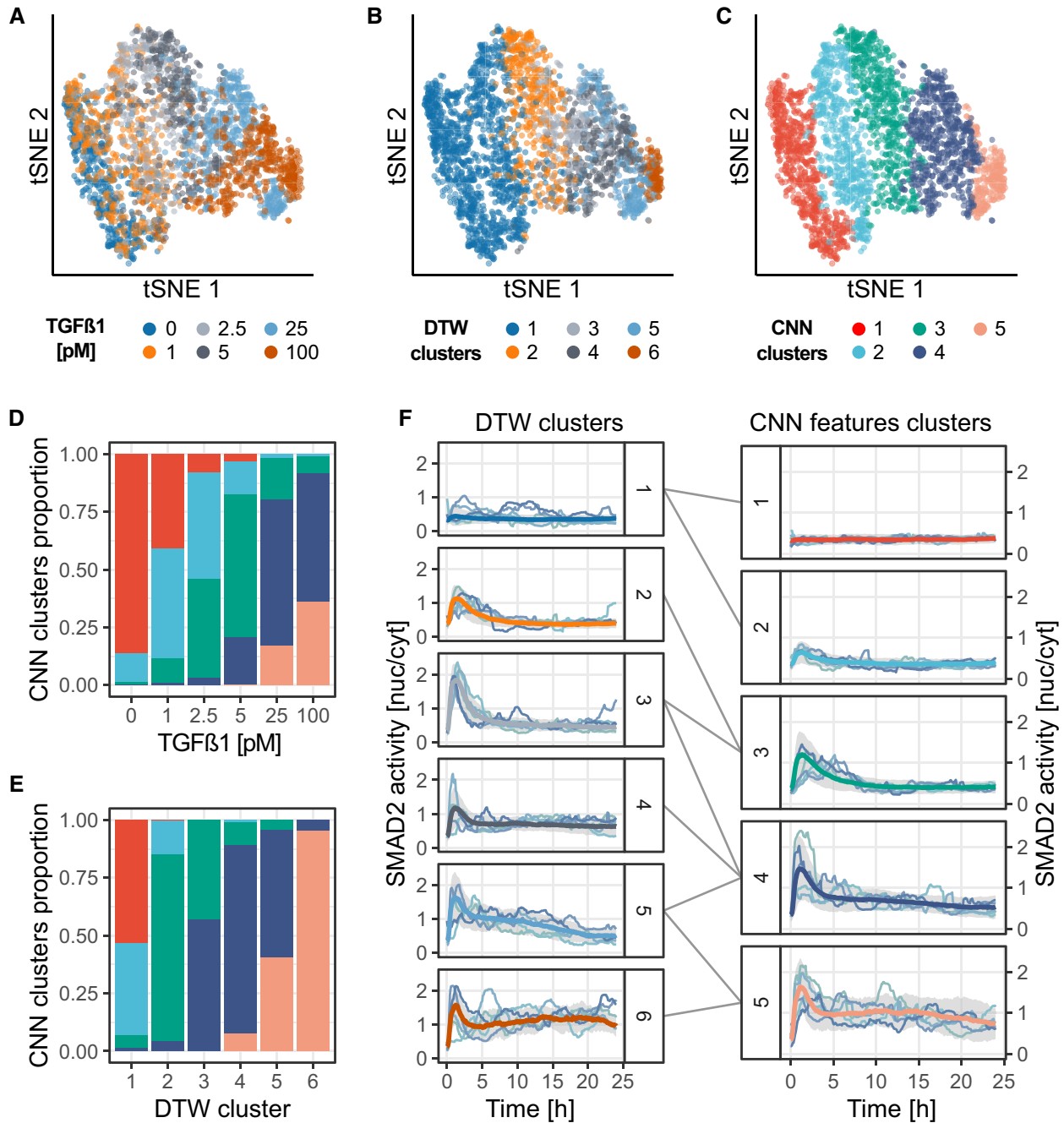

**Figure 4.  CODEX identifies TGFβ dose-dependent signaling states.**

MCF10A cells were exposed to increasing doses of TGFβ, and single-cell SMAD2 responses were recorded using a fluorescent biosensor (Strasen *et al*, 2018). A CNN was trained to classify SMAD2 activity trajectories according to the TGFβ dose.

A–C    t-SNE projection of the CNN latent features for the training, validation, and test sets pooled together. Trajectories representations are colored according to: the TGFβ dose (A), the DTW clusters (B), or the CNN features clusters (C). The DTW clusters are the ones of the original study (Strasen *et al*, 2018). CNN features were clustered using L1 distance and partitioned with hierarchical clustering and Ward linkage.

D, E    Distribution of the trajectories in the CNN features clusters according to their corresponding TGFβ dose (D) and their DTW cluster (E). Same colors as in (C).

F    Comparison of DTW clusters and CNN features clusters. Median trajectories for each cluster (see Materials and Methods) are reported in bold colored lines, colors matching (B) and (C), gray shade indicate interquartile range. The DTW clusters (1–6) are made of (946, 340, 200, 278, 218, 83) single-cell trajectories, respectively. The CNN clusters (1–5) are made of (505, 429, 451, 492, 188) single-cell trajectories, respectively. A random sample of trajectories for DTW clusters and the centroid trajectories (see Materials and Methods) for each CNN features cluster are shown.

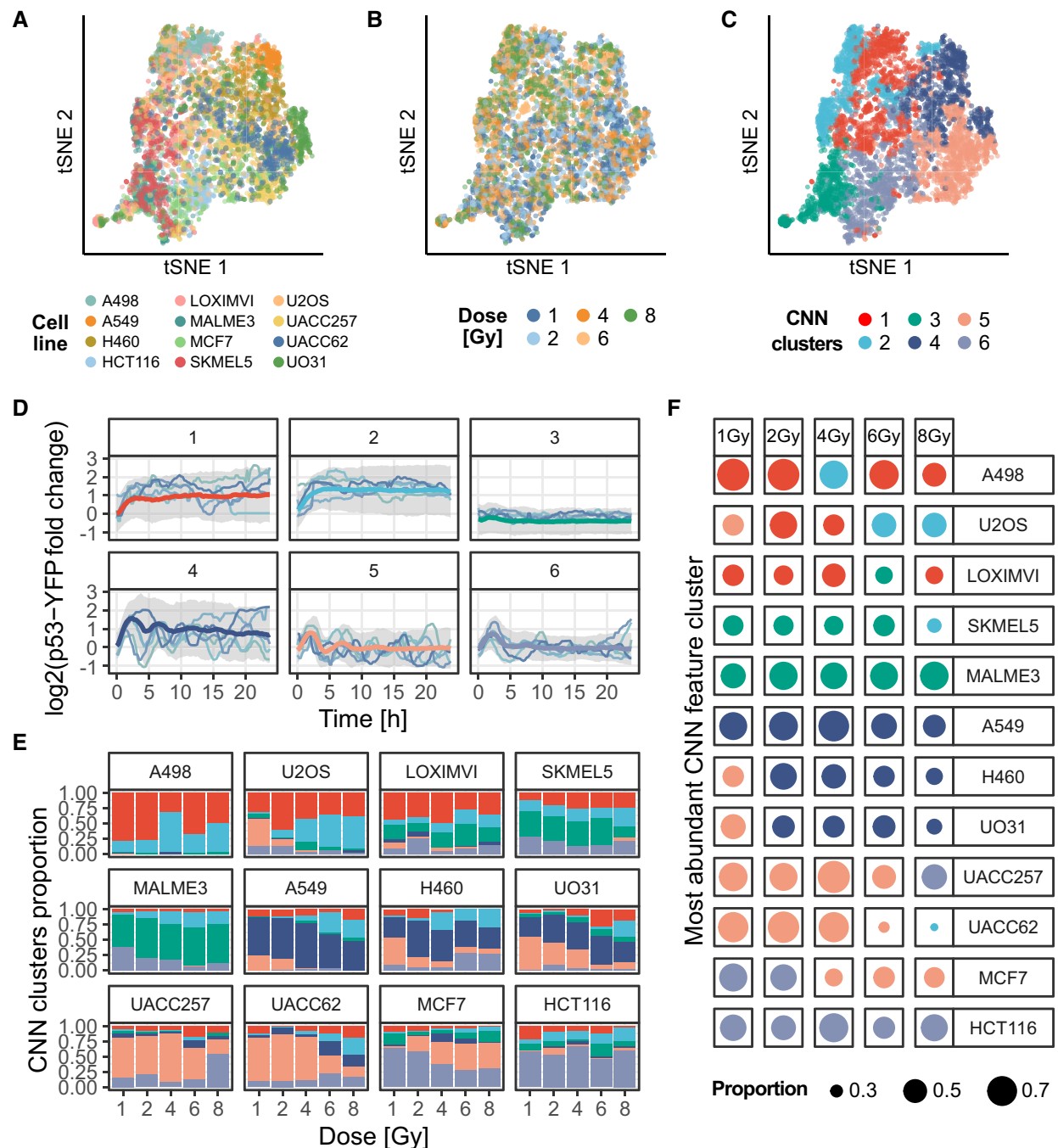

**Figure 5. CODEX identifies cell line-specific p53 responses under increasing ionizing radiation doses.**

12 different cell lines were exposed to five doses of ionizing radiations, and single-cell p53 responses were reported using a fluorescent reporter (Stewart-Ornstein & Lahav, 2017). A CNN was trained to recognize the combination of cell line and radiation dose from p53 trajectories (i.e., one out of 60 classes; see Materials and Methods and Appendix Note 4).

A–C   t-SNE projection of the CNN features for the training, validation, and sets pooled together. Trajectories representation is colored according to: the cell lines (A), the ionizing radiation doses (B), or the CNN features clusters (C). The p53 trajectories were clustered according to their standardized CNN features with L1 distance and partitioned with hierarchical clustering and Ward linkage.

D   Representative p53 trajectories of the clusters identified using CNN features clustering. Median trajectories are reported in bold colored lines, colors matching (C), gray shade indicates interquartile range. The CNN clusters (1–6) are made of (767, 554, 486, 574, 746, 698) single-cell trajectories, respectively. Data are pooled from two to three replicates for each cell line. Individual traces indicate the medoid p53 single-cell trajectories for each cluster (see Materials and Methods).

E   Distribution of the trajectories in the CNN feature clusters according to cell lines and ionizing radiation doses. Same colors as in (C).

F   Most abundant CNN features cluster for all trajectories in each combination of cell line and radiation dose combinations. The color of the dot indicates the most abundant cluster; the size of the dot indicates the proportion of cells classified in the indicated cluster. Same colors as in (C).

and applied it to our ERK and Akt trajectories. We then used these features for 2 data exploration workflows (Fig 3A). The first one is a supervised approach where we passed the features into a random forest classifier and inspected the feature importance for the classification (Fig 3B, Tables EV2 and EV3). This provided a rather incomplete overview of the dataset trends as many features are correlated or hard to interpret despite them being explicit.

In the second approach, we first projected the features with principal component analysis (PCA) and visualized the first components. This projection provided a rather entangled visualization of the dataset, as is commonly observed in high-dimensional datasets (Fig 3C–D). However, the interpretation of the features contributing to the three first components delineated rough but clear trends that are related to ERK variability, Akt variability, and Akt range of activity (Fig 3E). Finally, we sought to improve the projection, by first selecting enough components to cover 75% of the total variance and projected them with t-SNE (Fig 3F). We obtained a similar result to the projection of CODEX's data-driven features. Notably, the ErbB1 ligands (i.e., EGF, EPR, and BTC) are gathered and the IGF subpopulations are visible. However, HRG- and EPR-treated cells are largely scattered, whereas they show a remarkable compactness in CODEX projection. Visual inspection of individual trajectories suggests that this is because the features emphasize the range of the data, while CODEX puts more weight on discriminative patterns (Fig 3G). For feature-based methods, this reveals the bias that can be introduced by preselecting features. Although this preselection could also provide a clearer and/or targeted overview of the data, it also implies that some priors are needed about the data.

### CODEX is readily usable for a wide range of datasets and summarizes previous discoveries for other signaling pathways

To showcase the versatility of CODEX and benchmark it against classic approaches, we analyzed additional biosensor signaling datasets from previous studies. In one example, we analyzed SMAD2 dynamics in a TGFβ dose–response experiment in MCF10A cells (Strasen *et al*, 2018). Each TGFβ dose induced different heterogeneous SMAD2 dynamics that were isolated by shape-based clustering using DTW distance (Fig 4A, B and F). As the concentration of TGFβ increases, SMAD2 activity progressively shifts from low responses toward transient and then sustained activity. This change of activity was found to better correlate with the number of cell divisions and cell motility than the actual dose of TGFβ (Strasen *et al*, 2018). We were able to recover all dynamic trends highlighted by DTW with a clustering of the features of a CNN that was trained to recognize TGFβ doses from SMAD2 trajectories (Fig 4C–F, Appendix Note 3). CNN features even proved to be better at separating non-responders from low responders (Fig 4F, right panel, clusters 1 and 2). This further validates that CNN features are shaped after dynamics and that this can be used as an alternative to traditional shape-based clustering to capture biologically relevant dynamics.

In another example, we analyzed p53 dynamics in 12 different cancer cell lines subjected to five different doses of ionizing radiation (Stewart-Ornstein & Lahav, 2017). In this dataset, the combination of cell lines and radiation doses yields a total of 60 classes with dozens of p53 trajectories each (Fig 5A and B). The oscillations of p53 were described as a key signaling mechanism for regulating cell cycle arrest and apoptosis upon DNA damage (Lahav *et al*, 2004).

Here, the size of this dataset poses a major challenge for data visualization and mining. We trained a CNN to take p53 trajectories as inputs and recognize each of the 60 combinations of cell lines and radiation doses. A visual inspection of the CNN features projection revealed that the features separate better the cell lines than the radiation doses (Fig 5A and B). This hinted that cell line identity rather than radiation dose dictates p53 dynamics. In addition, the five melanoma cell lines of this dataset (LOXIMVI, MALME3, SKMEL5, UACC257, and UACC62) are scattered in very different areas of the projection. This suggests that p53 variation is not tissue-specific. Then, a clustering of CODEX features allowed us to rapidly identify discrete p53 signaling profiles (Fig 5C and D) and evaluate their distribution across conditions (Fig 5E and F). This recapitulated most cell line-specific effects that were described in the initial study (Appendix Note 4). For example, MCF7 cells uniquely showed increased p53 oscillations under high radiation doses. On the contrary, U2OS cells switched from a pulsatile regime under low radiation to more sustained response under high radiation. This analysis shows how CNN features can be used to visualize and analyze middle-size screens containing hundreds of asynchronous trajectories to extract biologically relevant insights.

Finally, we used CODEX on a time series dataset representing the movement speed of male and female *Drosophila melanogaster* under day and night light conditions (Fulcher & Jones, 2017) (Fig EV5A and B and Appendix Note 5). The trajectories in this dataset are significantly different in length and shapes compared to the other datasets and were not generated by biosensors. Despite this, with the same CNN architecture, the model converged to an excellent classifier, whose output correlates with discriminative, interpretable features (Fig EV5C and D) that were previously reported.

## Discussion

CODEX provides a new angle on extracting and mining features from large time series datasets. Instead of relying on user-defined features and their statistical significance, our approach learns features and uses them to highlight informative pieces of data. In our approach, a single model generates three views of the data: the projection, the prototypes, and the motifs, which can be explored interactively in the web application. For example, motifs revealed with CAMs in trajectory prototypes can be directly linked to a subpopulation of trajectories in the projection of the CNN features.

With CODEX, we were able to link the dynamics of ERK and Akt to biological aspects of the signaling networks. Strikingly, we found that BTC, EPR, and EGF, which are all ligands of ErbB1 and ErbB4 (Yarden & Pines, 2012), induced similar dynamics in both pathways. Indeed, the CNN often mislabeled one as another (Tables EV2 and EV3), they were clustered together in the low-dimensional projection (Fig 1B), and their prototypes revealed similar profiles (Fig 1C, Appendix Fig S1). Their common dynamics include a very correlated, pulsatile activity of ERK and Akt and high levels of ERK activity. The comparison of prototype trajectories showed that the three ligands differ mostly by the frequency of activity pulses. In stark contrast to these three ligands, HRG, a ligand of ErbB3 and ErbB4, transiently induced very high levels of Akt activity along with trains of ERK pulses. Altogether, these observations about signaling dynamics can help to understand various facets of the

signaling networks (Kholodenko *et al*, 2010; Ryu *et al*, 2015; Blum *et al*, 2019). First, the difference of pulse frequency across the ErbB1 ligands can help to understand the different kinetics of RTK dimerization upon various ligands binding. Then, despite ErbB4 being a common receptor for BTC, EPR, EGF, and HRG, the prototypes show very different activity linked to ErbB1 and ErbB3. This could shed light on how receptors from the same family, can finely modulate cell responses to external cues. Finally, the identification of minimal GF-dependent signaling motifs can provide a starting point for modeling the signaling networks. For example, in MCF10A cells, an excitable network generates pulses of ERK activity at random intervals (Albeck *et al*, 2013). Our finding that different growth factors induce ERK pulses of different shapes strongly suggests that different network feedbacks are triggered downstream of the different RTKs. The CAM-based motif mining lays the ground to formulate hypotheses and test mechanistic models that can recapitulate these striking dynamics (Ryu *et al*, 2015; Blum *et al*, 2019).

CODEX also successfully recapitulated previous findings for TGFβ/SMAD (Fig 4) and p53 (Fig 5) signaling through clustering of the CNN features. In the case of TGFβ/SMAD, we could retrieve previously reported dynamic profiles of SMAD2 signaling in response to increasing TGFβ doses (Strasen *et al*, 2018). These profiles correlated with cell division and cell motility and were key to delineate negative feedbacks in TGFβ signaling as well as sources of signaling heterogeneity between cells. In the case of p53, we summarized cell line-specific dynamics under increasing doses of ionizing radiation thanks to the clustering of the CNN features (Stewart-Ornstein & Lahav, 2017). The identification of such dynamics enabled to characterize variation in DNA repair efficiency and in the activity of the ATM kinase across cell lines. Such insights into signaling dynamics could be instrumental for the design of tailored treatments against any given tumor.

The shift of paradigm from user-defined to data-driven features relies on the capacity of the latter to capture complex relationships across spatial and temporal scales or between data from independent sources. This becomes increasingly relevant for biological applications such as the quantification of tissue properties from live cell imaging (Mergenthaler *et al*, 2021) or the integration of multi-omics experiments (Sharifi-Noghabi *et al*, 2019). In the context of cell signaling, CODEX can simultaneously mine the activity of multiple pathways, which enables to comprehensively study signaling dynamics and pathways crosstalk. For example, the motif induced by HRG, that comprises trains of ERK peaks enveloped by large Akt waves (Fig 2A and B), appears unambiguously in CODEX's output. By comparison, such motif would be hard to identify with general features, such as correlation, which would capture the motif very indirectly. Instead, CODEX provides a complete overview of the data thanks to the interplay between projection, prototypes, and motifs extraction. For example, CAM-based motifs can be overlaid directly on the prototypes or while browsing the projection of CNN features. This is a clear demarcation between CODEX and classic workflows because in the latter, motif mining constitutes a separate task which must be reconnected to other results afterward.

The use of data-driven features also removes the bias that can be introduced by preselecting features in the early stages of data exploration. However, a major drawback of data-driven approaches is that they are created through the optimization of an objective function which is itself predefined. Therefore, there is no guarantee that data-driven features will capture all interesting phenomena in the dataset, that the features will be interpretable, or that they will not be skewed by data artifacts (e.g., inter-replicate variance). Rather than competing, data- and feature-driven approaches are remarkably complementary. The former are great hypothesis generators because they can combine and summarize complex data. The latter are useful for checking whether the features were unbiased or correctly interpreted and are ultimately the only way to validate quantitative hypotheses. Regarding non-quantitative analysis, such as the motif extraction, it is to note that other robust methods exist to mine motifs in time series (Berndt & Clifford, 1994; Yeh *et al*, 2017). However, the CAM alternative is well-integrated with the other parts of CODEX.

Altogether, these results show that CODEX is a flexible framework, which can be applied to a wide range of data and robustly accommodate other model architectures (Fig EV3). This flexibility opens exciting perspectives for the extension of CODEX to new applications. For example, an interesting extension could be to study intercellular signaling by training a CNN that takes as input the signaling activity of a cell and of its neighbors simultaneously. Convolution operations on such data could therefore run both in time and space simultaneously. Such analysis could be valuable for the study of collective signaling events associated with collective cell migration (Aoki *et al*, 2017) or epithelium homeostasis (preprint: Gagliardi *et al*, 2020).

In summary, we have shown that CODEX provides a universal approach to quickly build hypotheses and identify phenotypes in dynamic signals from a wide variety of biological systems. Beyond this, CODEX demonstrates how CNNs, often criticized for their opacity, can reduce the workload of mining large datasets and suggest targeted, interpretable analysis.

# Materials and Methods

**Reagents and Tools table**

| Reagent/Resource | Reference or Source | Identifier or Catalog Number |
|---|---|---|
| **Experimental Models** | | |
| MCF10A (*Homo sapiens*) | Brugge laboratory | |
| **Recombinant DNA** | | |
| pMB-PB-FoxO3A-mNeonGreen | Pertz laboratory | |

**Reagents and Tools table**  (continued)

| Reagent/Resource | Reference or Source | Identifier or Catalog Number |
|---|---|---|
| pHygro-PB-ERK-KTR-mTurquoise2 | Pertz laboratory | |
| pPBbSr2-miRFP703 | Pertz laboratory | |
| **Chemicals, enzymes, and other reagents** | | |
| Recombinant Human IGF-I | PeproTech | 100-11 |
| Recombinant Human Betacellulin | PeproTech | 100-50 |
| Recombinant Human Epiregulin | PeproTech | 100-04 |
| Animal-Free Recombinant Human EGF | PeproTech | AF-100-15 |
| Human HGF | PeproTech | 100-39-10UG |
| Animal-Free Recombinant Human Heregulinβ-1 | PeproTech | AF-100-03 |
| Insulin solution human | Sigma-Aldrich | I9278-5ML |
| Hydrocortisone | Sigma-Aldrich | H0888-1G |
| Horse serum Donor Herd | Sigma-Aldrich | H1270-500ML |
| DMEM/F12 Ham | Sigma-Aldrich | D6434-500ML |
| Penicillin/Streptomycin | Sigma-Aldrich | P4333 |
| BSA | Sigma-Aldrich | A2153 |
| FuGENE® HD Transfection Reagent | Promega | E2311 |
| Hygromycin B solution | Santa Cruz biotechnology | sc-29067 |
| Blasticidin S HCl | Tocris Bioscience | 5502 |
| Puromycin dihydrochloride | Sigma-Aldrich | P7255 |
| **Software** | | |
| Ilastik | v1.3.2 | |
| Cell profiler | v3.1.8 | |
| MATLAB | R2016b | |
| **Other** | | |
| Eclipse Ti inverted fluorescence microscope | Nikon | |
| Zyla 4.2 plus camera | Andor | |

## Methods and Protocols

### Cell culture and biosensor imaging

MCF10A cells were cultured in DMEM:F12, 5% horse serum, 20 ng/ml recombinant hEGF (PeproTech), 10 µg/ml insulin (Sigma-Aldrich/Merck), 0.5 mg/ml hydrocortisone (Sigma-Aldrich/Merck), 200 U/ml penicillin, and 200 µg/ml streptomycin. GFs stimulation experiments were executed after 2 days' starvation in DMEM:F12, 0.3% BSA (Sigma-Aldrich/Merck), 0.5 mg/ml hydrocortisone (Sigma-Aldrich/Merck), 200 U/ml penicillin, and 200 µg/ml streptomycin. hEGF, BTC, EPR, HGF, HRG, and IGF (Pepro-Tech) were pre-diluted in the starving medium and added to cells under the microscope.

H2B-miRFP703, ERK-KTR-mTurquoise2, and FoxO3a-mNeon-Green constructs were generated and subcloned in the piggy PiggyBac vectors pMP-PB, pSB-HPB, and pPB3.0. Blast as previously described (preprint: Gagliardi et al, 2020). Upon transfection of these plasmids with FuGene (Promega), cells were treated with 2.5 µg/ml Puromycin, 25 µg/ml Hygromycin, and 5 µg/ml Blasticidin to select stably expressing cells. To achieve uniform biosensor experiments, cells were further cloned.

For imaging experiments, MCF10A cells were seeded on 5 µg/ml Fibronectin (PanReac AppliChem)—coated 24 well 1.5 glass bottom plates (Cellvis) at $1 \times 10^5$ cells/well density 2 days before the experiment. Time-lapse epifluorescence imaging was executed with an Eclipse Ti inverted fluorescence microscope (Nikon) equipped with a Plan Apo air 40× (NA 0.9) objective. Images were acquired with a 16-bit Andor Zyla 4.2 plus camera and with the following excitation and emission filters (Chroma): far red: 640 nm, ET705/72m; NeonGreen: 508 nm, ET605/52; mTurquoise2: 440 nm, HQ480/40. Images were acquired with 1,024 × 1,024 resolution with 2 × 2 binning.

### Automated image analysis

To obtain single-cell bivariate signaling trajectories of ERK and Akt activities, we used a dedicated image analysis pipeline, as previously described (preprint: Gagliardi et al, 2020). First, we trained a random forest classifier based on different pixel features with Ilastik (Berg et al, 2019) to separate H2B-miRFP703 fluorescence from background signal. The 16-bit nuclear probability channel produced by pixel classification was then used for nuclear segmentation with CellProfiler 3.0 (McQuin et al, 2018). A 7 pixels expansion of nuclear segmentation with two pixels separation was used to

produce a ring-shaped ROI in the cytosol. A ratio of the median pixel intensities of the cytosol over the nucleus masks was then calculated. Centroid-based single-cell tracking was executed with MATLAB using μ-track 2.2.1 (Jaqaman *et al*, 2008).

ERK and Akt activities were calculated as cytosol/nuclear (C/N) ratio of average pixel intensities in cytosol and nuclear ROIs in the respective fluorescence channels. Color coded images of ERK and Akt activities (Fig EV2B and C) were generated by color coding nuclear segmentation with the C/N values for each cell in each time point (CellProfiler 3.0).

ERK and Akt data analysis in single-cell trajectories was carried out with custom R codes. Heat maps of signaling trajectories (Fig EV2D) and average plus 95% confidence interval (Fig EV2E) were generated with Time Course Inspector (Dobrzyński *et al*, 2019).

### CNN architecture, training parameters, data augmentation, and preprocessing

All CNNs were built with the same convolutional architecture (preprint: Zhou *et al*, 2015), while only the number of filters in the last convolutional layer, i.e., the number of CNN features used for classification and projection, varied based on the dataset (Table EV1). For each dataset, several values for the number of CNN features were tested to minimize overfitting while maintaining predictive power (Appendix Note 1). The ResNet architecture (He *et al*, 2016) which was trained for the GF dataset is the only model with a different architecture. It starts with a layer of 2D-convolution with 20 kernels of size (3, 3); this is followed by three residual blocks with a skip connection regrouping two successive 2D-convolutions with 20 kernels of size (3, 3) each; a global average pooling generates a vector of features that are passed to a fully connected layer for classification. For all models, batch normalization and rectified linear units (ReLU) activation were used after each convolution layer. This standard setup was shown to improve training speeds and prevent overfitting (Krizhevsky *et al*, 2012; preprint: Ioffe & Szegedy, 2015). In this configuration, convolutions on bivariate data are done as if the signal were an image with two rows of pixels and a single-color channel. All models were trained to minimize cross-entropy loss with a L2 regularization weighted at 1e-3. Learning rates were initialized at 1e-2 and progressively reduced through epochs. The number of epochs varied from about 20 epochs for the synthetic set to a few hundred for the GF dataset, but all were trained in less than an hour on a consumer-grade GPU (Nvidia RTX 2080 Ti).

For all datasets, input trajectories were preprocessed before being passed to the CNN by subtracting from each channel its average value in the training set. The chosen CNN architecture imposes a fixed input size. We propose to take advantage of this limitation to perform data augmentation by randomly cropping trajectories before presenting them to the network (Table EV1). The procedure is analogous to what is commonly done on images, where cropping has been shown to be efficient at enforcing space- (in our case, time-) invariant feature learning (Krizhevsky *et al*, 2012). For the GF dataset analysis, we fixed a set of input trajectories to get rid of any variation due to the random crop before creating all figures related to these data.

### Prototype trajectories selection

We use the classification confidence of CNNs to identify prototype curves that are representative of the input classes. The classification

output of a CNN consists in a one-dimensional vector of real numbers, in which each number represents a single class. These numbers are not bound to a specific range but a higher number, relative to the rest of the output vector, is a stronger indication that the input belongs to a given class. As is usually performed, we transformed these output vectors with the softmax function. This squeezes all numbers between 0 and 1 and ensures that their sum is equal to 1. Hence, it gives a "probabilistic" interpretation to the CNN output. We define the classification confidence of a model for a given input, as the predicted probability for this input to belong to a given class.

We distinguish two types of prototype trajectories. On one hand, the "top prototypes" which are the trajectories for which the model prediction is correct and for which its confidence is the highest in a set of input trajectories. On the other hand, the "uncorrelated prototypes" which is a subset of input trajectories that we extract in two steps. First, the set of input trajectories is filtered to retain those for which the model confidence in the correct prediction reaches a predefined threshold. Second, a greedy algorithm chooses one by one trajectories in this filtered set such that their CNN features are as least correlated as possible between each other. This procedure is initiated by selecting the trajectory which has the highest median Pearson correlation to all the other trajectories in the filtered set.

For the GF dataset (Fig 1B and C), the 10 top prototypes from each class in the validation set were selected. For the synthetic dataset (Fig EV1C), eight uncorrelated prototypes with minimal confidence of 90% were chosen in the training and validation sets pooled together. For the drosophila movement dataset (Fig EV5B), the top prototypes for each class were chosen in the training and validation sets pooled together.

### Motif mining and clustering with CAMs

Class-discriminative motifs were identified with class activation maps (CAMs), a technique to reveal class-specific regions, according to a CNN classifier, in the input trajectories (preprint: Zhou *et al*, 2015). CAMs assign a quantitative value to each data point in input trajectories, large values indicate points that largely affect the model prediction toward a class of interest.

Here, we define a motif as a continuous stretch of points in the input trajectories that are recognized as important by the CNN for a given class, as indicated by CAMs. This approach is analogous to what was already proposed in computer vision (Selvaraju *et al*, 2017). To obtain the motifs, the points in an input trajectory are first classified as "relevant" or "non-relevant" for the class by binarizing CAM values using Li's minimum cross-entropy threshold (Li & Tam, 1998). This results in a collection of continuous "relevant" segments which are expanded by a defined number of points. This helps to better capture the context around a motif and to correct for single points detected as "non-relevant" by the thresholding. This collection of extended segments in a trajectory forms the collection of class-specific motifs in the trajectory.

In order to go beyond the identification of motifs among single trajectories, we established a motif mining pipeline to investigate and characterize motifs at the dataset scale. To do so, we first isolate motifs in prototype trajectories from every class in the datasets. The CAMs to identify these motifs are generated toward

the actual class of each trajectory. Then, from each trajectory only the longest motif is retained. Finally, all the motifs are compared with dynamic time warping (DTW) distance and partitioned with hierarchical clustering.

For the GF (resp. Synthetic) dataset, 75 (resp. 125) top prototype trajectories (resp. uncorrelated prototype trajectories with minimal confidence of 90%) were selected from both the training and the validation sets to extract the patterns. Motifs were extended by 2 (resp. 0) points, and only the longest motif was retained in each trajectory. The motifs were finally filtered to retain motifs shorter than 100 points (resp. longer than five points) (Figs 2A and EV1D).

### Dynamic time warping

Dynamic time warping (DTW) distances between CAM-motifs were computed with the R package *parallelDist*, with the "symmetric 2" step pattern. Pairwise distances were normalized by the sum of the lengths of both motifs of the pair.

### Medoids and centroids for motifs clusters and CNN features clusters

To present the content of the motif clusters (Figs 2A, EV1D and EV3C), sets of representative motifs were selected. The selection process explicitly relies on the distance matrices that were used to perform the clustering. Specifically, the medoid (resp. centroid) motifs are the motifs for which the median (resp. mean) DTW distance to all other motifs in the same clusters is minimal.

A similar procedure was followed to choose representative trajectories from CNN features clusters (Fig 4F right column, Fig 5 D). These clusters were defined by hierarchical clustering using the L1 distances between the CNN features of the trajectories. This distance matrix was used to select trajectories that minimized intra-cluster distance. For the TGFβ/SMAD2 dataset, centroids were used in place of medoids, i.e., trajectories that minimized the average intra-cluster distance (and not the median) were selected.

### t-SNE projections

The t-SNE projections of the CNN features were performed with the implementation in the Python library *sklearn*.

### Peak detection

The number of ERK/Akt activity peaks was calculated with a custom algorithm that detects local maxima in time series. A local maximum is defined as a point that exceeds the value of its neighbors by a threshold that was manually set at 0.12 for ERK and 0.10 for Akt.

### Classic time series features extraction, projection, and random forest classifier

Hundreds of classic time series features were extracted using the library *TSfresh* (Christ *et al*, 2018). Features were extracted both for ERK and Akt signals in the GF dataset. The features were extracted using the same fixed cut of data as the one that was used to produce the figures related to the CNN method (see CNN architecture method section). The features were filtered to keep only those that were significantly different between the GFs. This filtering was done using *TSfresh*'s procedure. The procedure runs

an appropriate statistical test for each feature and filters relevant features with a false discovery rate control of 5%. We then trained a set of random forests using the *H2O* R package (https://github.com/h2oai/h2o-3) that took these features as input and predicted the GF treatment. The parameters of the random forests were randomly sampled from predefined ranges and the model with best classification accuracy on the validation set was kept after 8h of search. This model comprised 386 trees with a maximum depth of split of 21; it reached a classification accuracy of 58%. The variables' importance of this model was extracted using *H2O*'s corresponding function. The latter estimates feature importance by the reduction of squared error associated with splits on each feature (Fig 3B).

The contributions of variables to each PCA component (Fig 3E) were extracted using the R package *factoextra* (https://github.com/kassambara/factoextra). The latter estimates a variable contribution to a PC by the ratio of its squared cosine over the sum squared cosines of all variables for a given PC.

### Synthetic data

Synthetic data were created by generating trajectories that always comprise 4 events of pulses (Fig EV1). Each pulse can be of two types: either a full Gaussian peak or a Gaussian peak truncated at its maximum. The side of the peak being truncated is random for each peak. The equation to generate a single peak event is as follows:

$$y_i(t) = H_i e^{\frac{-(t - P_i)^2}{(2S_i)^2}},$$

where $\{t | t \in N_0;\ t < 750\}$ is the discrete time variable, stopping when the desired length of trajectory is reached (here 750); $P$ is the discrete random variable for the time of event occurrence, follows $U\{0, 750\}$; $H$ is the height of the peak, follows $U(1, 1.5)$ and $S$ relates the width of the peak, follows $U(15, 25)$. After truncation of a peak, the final equation for a single peak is:

if no truncation:    if truncation to the left:    if truncation to the right:

$$y_i^*(t) = y_i(t),\ \forall t \quad y_i^*(t) = \begin{cases} y_i(t) & if\ t \geq P_i \\ 0 & otherwise \end{cases} \quad y_i^*(t) = \begin{cases} y_i(t) & if\ t \leq P_i \\ 0 & otherwise \end{cases}$$

The number of truncation events for a single trajectory is drawn from $U\{0, 1, 2\}$ for trajectories of the first class and $U\{2, 3, 4\}$ for trajectories of the second class. Finally, the whole trajectory is obtained by summing all independent peak trajectories:

$$z(t) = \sum_{i=0}^{4} y_i^*(t).$$

We add the hard constraint that each peak (i.e., each realization of $P$ in a trajectory) must be at least 75 time points away from each other.

The final synthetic dataset contains 10,000 trajectories in each class, from which 70% were used for training and the rest for validation.

## Data availability

All source data and code to reproduce every figure can be downloaded from: https://doi.org/10.17632/4vnndy59fp

The code necessary to run CODEX, along with user-friendly Jupyter notebooks and the interactive application to browse the t-SNE projection of CNN features are freely available at: https://github.com/pertzlab/CODEX.

**Expanded View** for this article is available online.

## Acknowledgements

The authors thank A. Loewer for providing the TGFβ/SMAD2 dataset, J. Stewart-Ornstein and G. Lahav for providing the p53 dataset, and J. van Unen for reviewing the manuscript. We acknowledge support by the Microscopy Imaging Center at the University of Bern (https://www.mic.unibe.ch/). This work was funded by an interdisciplinary (ID grant) of the University of Bern to OP and RS, and Div3 Swiss National Science Foundation (SNF) and Swiss Cancer League (Krebsliga Schweiz) grants to OP.

## Author contribution

M-AJ, MD, RS, and OP designed the study. M-AJ wrote the source code and analyzed the data. PAG performed biosensor imaging. M-AJ, OP, PAG, and MD wrote the manuscript. OP, MD, and RS supervised the project.

## Conflict of interest

The authors declare that they have no conflict of interest.

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
