## [Review Process File · Molecular Systems Biology]

CODEX, a neural network approach to explore signaling dynamics landscapes

Marc-Antoine Jacques, Maciej Dobrzyński, Paolo Armando Gagliardi, Raphael Sznitman, and Olivier Pertz

DOI: [10.15252/msb.202010026](https://doi.org/10.15252/msb.202010026)

Corresponding author(s): Olivier Pertz (olivier.pertz@izb.unibe.ch)

Review Timeline:

Submission Date:	25th Sep 20
Editorial Decision:	19th Nov 20
Revision Received:	4th Feb 21
Editorial Decision:	19th Feb 21
Revision Received:	1st Mar 21
Accepted:	3rd Mar 21

Editor: Maria Polychronidou

Transaction Report:

Thank you again for submitting your work to Molecular Systems Biology. I would like to apologise once again for the exceptional delay in getting back to you with a decision, which was due to the fact that after repeated reminders we still have not received the comments of reviewer #1. The reason why I have waited for them until now is that they repeatedly promised to deliver comments, and I felt that it would have been constructive to have a third opinion from an expert, especially on a new methodology. Nevertheless, to not delay the process any further, we have now decided to proceed with making a decision based on the two available reports. Overall, the reviewers acknowledge that the presented tool seems useful. However, they raise a series of concerns, which we would ask you to address in a major revision.

While reviewer #3 is overall supportive, reviewer #2 raises some important points which would need to be convincingly addressed. Without repeating all of this reviewer's points, we think that it is important to:

- Provide further support for the superiority of the approach compared to available methods.
- Perform some follow up analyses to more convincingly demonstrate the ability of the proposed approach to derive concrete biological insights.
- Make sure that the methodology is described in sufficient detail. While reviewer #3 did not have issues accessing and understanding the methodology, reviewer #2 did, and we would like to emphasize that the methodological details and workflow should be accessible to the broader audience of the journal and not only to specialists. This would increase the chances of wide adoption of the method. I would like to point out that there are no space limitations in our Methods format.

Please let me know in case you would like to discuss in further detail any of the issues raised. All issues raised by the referees would need to be satisfactorily addressed.

REFEREE REPORTS

Reviewer #2:

In their manuscript, Jaques et al. present a machine learning approach to extract human interpretable prototype time courses and characteristic motifs from cell signaling time course data. They apply their method to a novel dataset comprising thousands of bivariate Akt and Erk single-cell trajectories in response to stimulation with various growth factors. Specifically, they train a convolutional neural network (CNN) to classify time courses according to the nature of the applied extracellular ligand. From the trained model, they extract Class-Activation Maps (CAMs) to reconstruct and visualize recurrent time course characteristics (which they call motifs). Thereby, they assign characteristic bivariate Akt and Erk dynamics to each growth factor which may hint towards underlying ligand-specific signaling mechanisms and/or network motifs. This approach generally demonstrates how the "black box" of NN can be opened up to gain interpretable biological insights and is of general interest in the analysis of single-cell time courses. However, in its present form, the manuscript has several weaknesses, as the description of the framework remains vague, the superiority of their approach as well as its applicability to gain concrete biological insights remain unclear.

Specific comments

- 1) In the second paragraph of the paper, the authors describe their CODEX approach and its application to a synthetic benchmark dataset. For a methods article, this description is very short, so that the main ideas underlying the framework and the benchmark are hard to follow. Despite space limitations, the authors should spend more efforts to describe the methodological concepts.
- 2) Then, the authors apply their method to a comprehensive live-cell imaging dataset comprising thousands of single-cell measurements of Akt and Erk dynamics in response to five different growth factors. Gaining interpretable insights into the temporal signaling dynamics in such a big dataset is a challenge, but could lead to highly relevant insights into growth factor specificity. The authors address this problem by training a neural network (NN), and extract of prototype time courses for each ligand by clustering of the NN features (Fig. 1B/C). The authors propose the one particular CNN architecture, but do not clearly motivate the chosen architecture or compare their approach against alternative models. For instance, they may compare their model against standard architecture, such as ResNet. As an alternative to the CAM method, the latent space

representation of an autoencoder might be applied (for instance, <https://arxiv.org/abs/1610.04794>).

3) Based on the CNN model, recurrent time course characteristics (motifs) are extracted using CAMs. Thereby, bivariate Akt and Erk dynamics of variable length are assigned to each growth factor (Fig. 2A/B). To us, this derivation of human interpretable time course characteristics for each ligand seemed to be the main novelty of the CODEX approach, but this is not clearly stated in the manuscript (see also comment 4). Furthermore, it is not directly apparent how such motifs will lead to biological insight. The authors write that they may be used for modeling of signaling networks. They should be more concrete about this statement, provide a specific example for such a modeling approach and discuss how modeling the motifs will be beneficial when compared to modeling the prototype time courses.

4) In relation to the previous comment, the authors should better highlight the novelties and advantages of their approach compared to existing methods.

5) In Fig. 2C, the authors provide combinations of lumped Akt and Erk signaling features (ratio and correlation) which best discriminate between the ligands. They should better explain what they mean when writing they were "bringing together the results from the three components of CODEX" to derive this. Is this a customized analysis for their specific dataset or can this approach be generalized to other biological systems?

Reviewer #3:

This short manuscript presents a software tool for analyzing patterns found in dynamic time course data. The tool uses a convolutional neural network machine learning algorithm to classify kinetic features within a many-sample dataset, and is particularly geared to deal with data from live-cell reporter experiments. The main analysis presented focuses on a dataset with two reporters (for ERK and Akt), in cells responding to a panel of growth factors that activate both pathways. The data for analysis were generated anew in this study, although they mirror a previously published study (Sampattavanich). I think this choice is a good one, as they are able to show that their analysis recapitulates much of what was found in the earlier paper, in some cases with even greater clarity, as well as some new features, all of which demonstrates the robustness and insight achievable with their tool. The authors also analyze additional published datasets, for SMAD signaling, p53, and drosophila sleep/wake patterns, with similarly clear results.

Overall, this work is presented in an extraordinarily clear way, and the software tool meets an important and growing need in the live-cell field. The authors describe a number of carefully considered choices that appear to make their tool both versatile and relatively easy to use. I think quite a few people working with live-cell data will find this tool extremely useful.

We were able to run the software without a problem, and we found it to be well documented.

Given the clear and quite thorough presentation, we could not find any serious issues to be addressed. The one area that could perhaps be improved would be some guidance for potential users in selecting the number of features in their CNN, which as they note is the only parameter that was adjusted between the different datasets; it would be useful to have a sense of what criteria are used when deciding how to set this value.

Point-by-point answers**Reviewer #1:**

Understanding how the dynamics of signaling proteins underpins different fate decisions at the cell, tissue, and organism level is a central question in cell biology. Over the last two decades a number of reporter systems have been developed to image these dynamics in single cells. By imaging these reporters over time, it is possible to generate signaling trajectories, which describe the dynamics of a particular reporter, and thus of a particular signaling event such as phosphorylation or nuclear translocation, in large populations of cells. However, due to the highly heterogeneous nature of these trajectories both within cells and between cells, quantifying and classifying these complex dynamics is challenging. Specifically, it is not always clear as to which aspect of a signaling trajectory is meaningful statistically or biologically.

Here Pertz and colleagues present CODEX, a data-driven approach that learns specific distinguishing features from different signaling trajectories. By then classifying different trajectories in an unbiased fashion (ie by projecting into t-SNE spaces), this starting point for understanding how these dynamic behaviours dictate fate decisions.

Packages like this are desperately required, and regardless of CODEX's particular efficacy, I think this work has impact. I wouldn't say anything here is particularly novel in terms of dynamic imaging analysis - much more sophisticated work has been done - but this methodology is very intuitive and user friendly. I can easily imagine cell biologists whose image cells expressing these reporters, to analyse their datasets. I have already started exploring its capabilities.

Instead of basing their analysis around user-defined features and their statistical significance, their approach learns features and uses them to highlight informative pieces of data. I like the intuition of using CNN's feature extraction capabilities here, it makes perfect sense. Moreover, CODEX provides a universal approach to quickly build hypotheses and identify phenotypes in dynamic signals from a wide variety of biological systems.

CODEX demonstrates how modern machine learning models, often criticized for their opacity, can reduce the workload of mining large datasets and suggest targeted, interpretable analysis.

Perhaps most importantly, they provide very well documented code and very easy to re-implement.

The authors provide a number of test cases for CODEX which I think demonstrate its utility. These are (but not limited to) the following:

- 1) On the synthetic dataset, the t-SNE projection of the CNN features learnt for this task shows well-separated trajectory clusters, which are grouping together trajectories with a common number of full peaks rather than a common class label. This is interesting as even though the model is only optimised to minimize the classification loss between the two classes (full and half peaks), clusters containing 0 or 1 full peak and the clusters containing 3 and 4 full peaks could be identified without affecting the classification performance. That the model learns specifications within each class show that CNNs can naturally evolve to capture shapes in the data even without hard constraints.

- 2) When projected into t-SNE space, growth factor (GF) induced trajectories populated well-defined but also slightly overlapping areas in the projection space. This suggests that each GF induces a specific continuum of heterogeneous signaling dynamics, whose characteristics correlate with their cognate GF receptor. I find this section very interesting - to me it does show the potential of CODEX to classify different complex responses.

- 3) Used DTW shape-based clustering to evaluate their distribution in different GF classes, the authors were able to provide additional insight about GF-specific

signalling dynamics. Specifically, the ErbB1 ligands BTC, EGF, EPR all led to a mix of synchronous ERK/AKT pulses in which AKT amplitude was lower than ERK amplitude, or wider ERK/AKT pulses with a very low AKT amplitude. HRG led to a peculiar pattern consisting of multiple sharp ERK pulses under larger AKT pulses. IGF led to sustained AKT, and baseline ERK activity with occasional pulses. Taken together CODEX results suggested that the synchrony between ERK and AKT as well as their amplitude ratio were highly discriminative across GF stimulations.

4) In a study of SMAD signaling dynamics, visual inspection of representative trajectories suggests that the CNN features performed slightly better in separating flat from weak responders in comparison with the DTW clusters.

5) CODEX allowed rapid identification of discrete p53 signalling profiles and evaluation of their distribution across conditions. CODEX could recapitulate important findings in a large time-series dataset with very little human input and in about one hour for training the model.

I really only have one major technical issue. The performance of the CNNs on actual dataset of ERK dynamics after stimulation is not impressive. The classifier separated the different classes with about 65% accuracy. While the authors claim that this "suggests input trajectories carry distinctive features that depend on GF identity many hours after exposure"; this accuracy isn't really compared to anything. Furthermore, when looking at the other metrics in the table, the precision, recall and f1-score of some treatments is not good at all. Whether this is inherent to the data itself, or whether the training could be improved, is slightly unclear at this point. This performance I think is something to consider on when judging the overall utility of this methods. I would like to hear from the authors on this.

With regard to the last issue, we agree that this classification performance could appear rather low in comparison to the very high performance that made the fame of CNNs. We believe there are multiple sources behind this. First, we intentionally limit our analysis to the "difficult" part of the data such that we are avoiding the early, strong and synchronized responses after GF addition. This is because we wanted to demonstrate how CODEX can help in a complex scenario where dynamics are hard to analyze but biologically-relevant. Second, we have chosen a CNN architecture that has a good interpretability and that is easy to train, but that is not geared for state-of-the-art classification performance. Classification performance might be improved with a more complex architecture, but it was not our main aim. Last, we also believe that the single-cell responses are highly heterogeneous, with features that are partially overlapping between the growth factors. These heterogeneity and overlap are such that the optimal classification accuracy is probably rather low. However, this heterogeneity results from noise in the biochemical networks, is biologically-relevant, and is an aspect of signaling networks that we wanted to capture.

Further, we specifically addressed this concern with 2 complementary approaches. First, we took a complete orthogonal approach by extracting hundreds of classic time-series features from ERK and Akt traces. We then used these features to train a random forest to recognize the GFs (*Figure 3 and pages 9*). The resulting model is performing significantly worse than its CNN counterpart (*Tables EV2, EV3*). This is to us, an indication that the low classification performance is inherent to the data and not to our analysis. Second, we show that a different CNN architecture (ResNet) is also struggling to separate GFs (*Figure EV3 and page 8, 2nd paragraph*). In fact, its classification accuracy is even slightly lower than the ones of the plain CNN (*Tables EV2, EV3*). Altogether this suggests that there is not much room for improvement of classification performance.

Minor issues:

As far as I understand it they claim that all CNNs are fully-convolutional, but in code they have fully-connected layers for the classifiers. Maybe, what they mean is that the feature extractors used remove these fully-connected layers in and only use the fully-convolutional part of the CNN?

By fully-convolutional we meant an architecture without intermediate pooling layers. The term was indeed misleading, hence we removed it.

The authors state that "only the number of filters in the last convolutional layer, i.e. the number of CNN features used for classification and projection, varied based on the dataset used", but don't give information on why specific feature sizes were chosen. Might be interesting to see what lead them to decision. Maybe just trial and error?

It was indeed a trial-and-error process guided by overfitting reduction and preservation of predictive power. For each model we tried to change a variety of parameters but found the number of CNN features to be the fastest and easiest to tune since it has a large and predictable effect (*i.e.* less features reduce overfitting). We sought to offer a solution in which as little time as possible is spent on model tuning and found that this a good trade-off to propose to new users. We do not exclude that users should try to tune other parameters, such as the L2 penalty, and make it easy to do so in our implementation. However, the changes are more subtle and we fear that the growing number of parameters to adjust could discourage potential users, while default options appeared robust in our testing. We have discussed this point more precisely in the main text (*page 4, Results section*), the method section (*page 13, 1st paragraph of the corresponding section*) and the appendix notes (*appendix Note 1, last paragraph*).

I am not sure I find the study of *Drosophila* behaviour particularly useful here.

This dataset was introduced to demonstrate the applicability of CODEX to time-series beyond cell signaling. We believe it was important to show a concrete application on data that were not generated by biosensors. In addition, we wanted to show that the same CNN architecture can work for trajectories with distinct profiles: biosensor data all vary rather smoothly whereas *Drosophila* trajectories are bursty. This last point supports the idea that CODEX can be quickly adapted to a range of datasets without heavy parameter tuning.

Reviewer #2:

In their manuscript, Jaques et al. present a machine learning approach to extract human interpretable prototype time courses and characteristic motifs from cell signaling time course data. They apply their method to a novel dataset comprising thousands of bivariate Akt and Erk single-cell trajectories in response to stimulation with various growth factors. Specifically, they train a convolutional neural network (CNN) to classify time courses according to the nature of the applied extracellular ligand. From the trained model, they extract Class-Activation Maps (CAMs) to reconstruct and visualize recurrent time course characteristics (which they call motifs). Thereby, they assign characteristic bivariate Akt and Erk dynamics to each growth factor which may hint towards underlying ligand-specific signaling mechanisms and/or network motifs. This approach generally demonstrates how the "black box" of NN can be opened up to gain interpretable biological insights and is of general interest in the analysis of single-cell time courses. However, in its present form, the manuscript has several weaknesses, as the description of the framework remains vague, the superiority of their approach as well as its applicability to gain concrete biological insights remain unclear.

Specific comments

1) In the second paragraph of the paper, the authors describe their CODEX approach and its application to a synthetic benchmark dataset. For a methods article, this description is very short, so that the main ideas underlying the framework and the benchmark are hard to follow. Despite space limitations, the authors should spend more efforts to describe the methodological concepts.

We fully agree with this comment. We were initially intending to release a very condensed manuscript using the growth factor dataset as a guiding example. However, under the light of the reviewers' comments, it appeared that some sections had to be largely expanded for intelligibility. In the present revision, the synthetic data

generation and the interpretation of the corresponding results were detailed at greater length in the main text (*page 5, 1st paragraph*).

2) Then, the authors apply their method to a comprehensive live-cell imaging dataset comprising thousands of single-cell measurements of Akt and Erk dynamics in response to five different growth factors. Gaining interpretable insights into the temporal signaling dynamics in such a big dataset is a challenge, but could lead to highly relevant insights into growth factor specificity. The authors address this problem by training a neural network (NN), and extract of prototype time courses for each ligand by clustering of the NN features (Fig. 1B/C). The authors propose the one particular CNN architecture, but do not clearly motivate the chosen architecture or compare their approach against alternative models. For instance, they may compare their model against standard architecture, such as ResNet. As an alternative to the CAM method, the latent space representation of an autoencoder might be applied (for instance, <https://arxiv.org/abs/1610.04794>).

We thank the reviewer for pointing out the lack of clarity on this point. We respectfully disagree with our absence of justification about the architecture choice because the appendix note 1 is covering this exact point. We do however agree that further explanations were needed in the main text. We have added such explanations (*page 4, first paragraph of the Results and Discussion section*), which should make the rationale clearer and guide new users aiming to apply the method.

We also agree that the absence of comparison to alternative models was problematic. To address this issue, we have redone the CODEX analysis for ERK and Akt activity with a different architecture. As was suggested, we used a standard ResNet architecture (He *et al*, 2016). Interestingly, we found the results to be very similar to the ones obtained with a plain CNN: the dataset projection, class prototypes and class-specific motifs are comparable (*Figure EV3 and corresponding text on page 8, 2nd paragraph; Tables EV2, EV3*). We believe that this experiment consolidates our previous findings about ERK and Akt dynamics and emphasizes the robustness of the analysis.

With regards to the Deep Clustering Network, we did not fully understand what was meant by an “alternative to the CAM method”. We agree that creating “clustering-friendly” features with an autoencoder can definitely be of interest to obtain a low-dimensional embedding of the dataset. However, this would come at the cost of dropping the prototype extraction and motifs extraction without further training of a classifier with the encoder features.

3) Based on the CNN model, recurrent time course characteristics (motifs) are extracted using CAMs. Thereby, bivariate Akt and Erk dynamics of variable length are

assigned to each growth factor (Fig. 2A/B). To us, this derivation of human interpretable time course characteristics for each ligand seemed to be the main novelty of the CODEX approach, but this is not clearly stated in the manuscript (see also comment 4). Furthermore, it is not directly apparent how such motifs will lead to biological insight. The authors write that they may be used for modeling of signaling networks. They should be more concrete about this statement, provide a specific example for such a modeling approach and discuss how modeling the motifs will be beneficial when compared to modeling the prototype time courses.

Following this suggestion, we have emphasized the novelty of CAM-based mining and its integration with the other results of CODEX (*page 9, last paragraph*). We agree with the reviewer that our very short claim was misleading, and not well documented. In no case did we mean that understanding signaling motifs can lead to a generalizable modelling framework to understand the network structure that generates these signaling dynamics. Rather than being a modelling objective, motifs help to pinpoint behaviors of interest which might not be immediately clear, even in prototype trajectories. We have now provided a concrete example on how the motif information could be used for modelling (*page 8, 1st paragraph*).

4) In relation to the previous comment, the authors should better highlight the novelties and advantages of their approach compared to existing methods.

We have addressed this issue with various edits that emphasize CODEX's "all-under-one-roof" nature, its focus on dynamical patterns, the ability to switch and combine the different analysis (*e.g. visualizing patterns on prototype trajectories*) and its ability to deal with data where one has no prior knowledge (*standfirst text; introduction, last sentence; page 9 last paragraph*).

In order to provide concrete material for such comparison, we have added an experiment where we analyzed the ERK/Akt dataset with classic machine-learning approaches (*Figure 3 and page 9*). To do so, we extracted hundreds of features from the trajectories and used them to explore the trends in the data by: 1. training a random forest classifier and inspect feature importance; 2. projecting the features with PCA. We found converging results between the classic approaches and CODEX but also advantages for the latter. Namely, although the time-series features are explicit, the variable importance of the classifier and the interpretation of the PCA components gives a very coarse-grained overview of the dataset. This classic "feature-first" paradigm lacks the resolution and the intuition that CODEX provides by highlighting selective pieces of data.

5) In Fig. 2C, the authors provide combinations of lumped Akt and Erk signaling features (ratio and correlation) which best discriminate between the ligands. They should better explain what they mean when writing they were "bringing together the

results from the three components of CODEX" to derive this. Is this a customized analysis for their specific dataset or can this approach be generalized to other biological systems?

This combination of features is an attempt to concisely summarize most insights that were acquired on this particular dataset thanks to a CODEX-based exploration. In that sense, this is indeed an analysis which is specific to this dataset. But the rationale of deriving targeted, quantitative measures following CODEX exploration is general. We have modified the text to clarify what is specific to this dataset and what is generalizable (*page 8, last paragraph*).

Reviewer #3:

This short manuscript presents a software tool for analyzing patterns found in dynamic time course data. The tool uses a convolutional neural network machine learning algorithm to classify kinetic features within a many-sample dataset, and is particularly geared to deal with data from live-cell reporter experiments. The main analysis presented focuses on a dataset with two reporters (for ERK and Akt), in cells responding to a panel of growth factors that activate both pathways. The data for analysis were generated anew in this study, although they mirror a previously published study (Sampattavanich). I think this choice is a good one, as they are able to show that their analysis recapitulates much of what was found in the earlier paper, in some cases with even greater clarity, as well as some new features, all of which demonstrates the robustness and insight achievable with their tool. The authors also analyze additional published datasets, for SMAD signaling, p53, and drosophila sleep/wake patterns, with similarly clear results.

Overall, this work is presented in an extraordinarily clear way, and the software tool meets an important and growing need in the live-cell field. The authors describe a number of carefully considered choices that appear to make their tool both versatile and relatively easy to use. I think quite a few people working with live-cell data will find this tool extremely useful.

We were able to run the software without a problem, and we found it to be well documented.

Given the clear and quite thorough presentation, we could not find any serious issues to be addressed. The one area that could perhaps be improved would be some guidance for potential users in selecting the number of features in their CNN, which as they note is the only parameter that was adjusted between the different datasets; it would be useful to have a sense of what criteria are used when deciding how to set this value.

The concern about explaining how to choose the number of CNN features is a very good point and will help for a wider adoption of the method. This parameter is essentially tuned with successive trials, with the objective to balance overfitting and predictive power. We added additional indications in the main text (*page 4, Results section*), the method section (*page 13, 1st paragraph of the corresponding section*) and the supplementary notes (*appendix Note 1, last paragraph*) to clarify this and guide new users.

Thank you again for sending us your revised manuscript. We have now heard back from reviewer #2 who was asked to evaluate your revised study. As you will see below, the reviewer is satisfied with the modifications made and thinks that the study is now suitable for publication.

Before we can formally accept the manuscript for publication, we would ask you to address a few remaining editorial issues listed below.

REFEREE REPORTS

Reviewer #2:

The authors comprehensively addressed all our comments. We congratulate them for their great work!

The authors have made all requested editorial changes.

Thank you again for sending us your revised manuscript and for performing the requested minor changes. I am pleased to inform you that your paper has been accepted for publication.

YOU MUST COMPLETE ALL CELLS WITH A PINK BACKGROUND

Corresponding Author Name: **Ulrich Pfaffl**
 Journal Submitted to: **Medical Systems Biology**
 Manuscript Number: **MS-20-1127**